# Optimization of the Physical Properties of HDPE/PU Blends through Improved Compatibility and Electron Beam Crosslinking

**DOI:** 10.3390/polym14173607

**Published:** 2022-09-01

**Authors:** Jin-Oh Jeong, Yong-Hyeon Oh, Sung-In Jeong, Jong-Seok Park

**Affiliations:** 1Wake Forest Institute for Regenerative Medicine (WFIRM), Wake Forest School of Medicine, Winston-Salem, NC 27157, USA; 2Advanced Radiation Technology Institute, Korea Atomic Energy Research Institute (KAERI), Jeongeup-si 56212, Korea

**Keywords:** polymer blending, compatibilizer, crosslinking, electron beam, composites

## Abstract

Polymer blending is a method in which polymers with different properties are mixed so that each advantage appears in one polymer blend. Improved thermal and mechanical properties of blends can be prepared by blending with high-density polyethylene (HDPE) of a non-polar polymer and polyurethane (PU) of a polar polymer. However, a compatibilizer is required because it has the disadvantage that blending has low miscibility due to the different phases. In this study, HDPE/PU blends with new and excellent physical properties were developed through optimal composition with improved compatibility between the HDPE and PU. In addition, the effects of improving the physical properties through electron-beam crosslinking were confirmed. In general, a crosslinking structure of HDPE is formed by electron beam irradiation to increase its thermal stability and strength, but its elongation is rapidly decreased. In particular, the elongation of HDPE irradiated at 100 kGy was about 110%, which was decreased about five times compared to unirradiated HDPE (510%). However, the HDPE/PU blend with improved compatibility (PU 30) showed an elongation of about 450% while maintaining excellent strength (22.5 MPa), which was increased by about four times compared to the HDPE irradiated at 100 kGy. In particular, the thermal stability of PU 30 irradiated at 100 kGy at a high temperature (180 °C) was improved more than six times compared to the HDPE. Therefore, it is possible to develop HDPE/PU blends with new and excellent physical properties by improving compatibility and using electron beam crosslinking technology.

## 1. Introduction

Polyolefin-based polymer resins are widely used in daily life because they have excellent mechanical strength, chemical resistance, and processability; however, these polymers have low thermal stability, impact strength, flame retardancy, and cold resistance. Polymer blending technology has been widely used to overcome these weaknesses of the homopolymer [1,2]. Polymer blending is a technology that enables the unique characteristics of each polymer to be imparted to one material through blending [3,4]. There are two types of polymer blending methods (chemical and physical methods) [5]. The chemical method is polymerization from a monomer and/or polymer copolymerization [6,7]. The physical method is that polymers with different physical properties can be blended through temperature and/or solution, regardless of the molecular weight of the polymer or the chain number [8]. The physical method has the advantage of being more intuitive and easily accessible compared to the chemical method. For example, when manufacturing a hard plastic that is not easily broken by impact, a composite can be prepared by blending composites with a glassy and elastic polymer in the desired polymer ratio using heat [9,10]. Thus, polymers widely used in polymer blending include polypropylene (PP), polystyrene (PS), polyurethane (PU), and polyethylene (PE) [11,12,13].

PE is a polymer to which a large number of ethylene monomers are bound and is classified into low-density polyethylene (LDPE) or high-density polyethylene (HDPE) according to the number and/or length of bound ethylene monomers [14,15]. HDPE is a translucent crystalline polymer with a high crystallinity of about 60–80%, a melting point of 135 °C, a density of 940 kg/m^3^, and a specific gravity of 0.94–0.96 [16]. Because HDPE is strong against impact due to its high crystallinity, it is widely used in various containers, plastic boxes, and water pipes [17,18,19]; however, the fluidity and processability of HDPE are limited due to a lower strength compared to PP, a weak heat resistance, and high shrinkage during molding [20,21]. To compensate for the weakness of HDPE, we previously prepared HDPE composites by the physical blending method with PU and EVA, which has an excellent heat resistance and processability. In addition, the physical properties of HDPE/EVA/PU composites were improved through polymer crosslinking using an electron beam. The crosslinked HDPE/EVA/PU blends showed excellent thermal shrinkage and enhanced mechanical properties [22].

The polymer crosslinking technique can improve and/or maintain the physical, thermal, and mechanical properties of composites [23,24,25,26]. In particular, because electron-beam-induced crosslinking has the characteristic of ionizing a material by itself, it can induce a reaction without chemical additives (e.g., crosslinking agent, initiator, or catalyst) and can be induced in various environments (e.g., oxygen, nitrogen, and vacuum) and conditions (e.g., solid and liquid), which are the advantages for using an electron beam [27,28,29]. When HDPE is treated with electron-beam irradiation, the hydrogen of the tertiary carbon atom of HDPE disappears, and a crosslinking reaction occurs to stabilize the free radicals created with each other [22]. Accordingly, polymers can be crosslinked through electron-beam irradiation without chemical additives.

In this study, PE/PU blends with new and excellent physical properties were developed through optimal composition with improved compatibility between PE and PU. In addition, the effects of improving the physical properties through electron-beam crosslinking were confirmed. Due to the improved compatibility and electron-beam crosslinking technology, the HDPE/PU blends satisfied the strength and elongation requirements at the same time, and their high thermal stability was also improved by more than four times compared to pure HDPE. These excellent properties of the PE/PU blends, through improved compatibility and electron-beam crosslinking, can be used in insulated cables, pipes, and building materials that have to withstand harsh external environments.

## 2. Materials and Methods

### 2.1. Materials

High-density polyethylene (HDPE) was purchased from the Lotte Chemical Corporation (Seoul, Korea). Polyurethane (PU) was obtained from Songwon Industrial Co., Ltd. (Ulsan, Korea). Polyethylene-graft-maleic anhydride (PE-g-MA) (viscosity of 500 cP) was purchased from Sigma-Aldrich (St. Louis, MO, USA). In addition, trimethylolpropane trimethacrylate (TMPTMA) was obtained from Tokyo Chemical Industry Co., Ltd. (Nihonbashi-honcho, Japan). All other reagents and solvents were of analytical grade and used as received.

### 2.2. Preparation of the HDPE/PU Blends

To prepare HDPE/PU blends, HDPE and PU were blended using a Brabender mixer (Brabender D-47055, Brabender, Duisburg, Germany) at 45 rpm and 190 °C for 20 min with different contents of PE-g-MA and TMPTMA, its compatibilizer. Sheets of the HDPE/PU blends (thickness of 2 mm) were prepared using a hot press at 190 °C. To achieve the crosslinking, the sheets were exposed to electron beam irradiation (2.5 MeV, UELV-10-105, Korea Atomic Energy Research Institute, Jeongeup, Korea) at radiation doses of 50, 75, and 100 kGy (25 kGy/cycle). The total chemical compositions are shown in Table 1 and Table 2.

### 2.3. Characterization of the HDPE/PU Blends

The elemental component analysis of the HDPE/PU blends was performed with energy-dispersive X-ray spectroscopy (EDS, TM3030, HITACHI, Tokyo, Japan). The samples were coated with gold for 60 s using a sputter coating observed with a 15 kV electron beam and a working distance of 8.1 mm.

The gel fraction of the HDPE/PU blends was measured with the Soxhlet extraction. The initial weight of the samples was recorded, then immersed into xylene at 140 °C for 8 h and finally dried in an oven at 80 °C for 8 h. Then, the samples were dried naturally for 4 h to record their weight. The gel fraction was calculated with the following equation:Gel fraction (%) = (W_f_/W_i_) × 100,
where W_i_ and W_f_ represent the initial and final weights of the dried samples, respectively.

To measure the tensile strength of the HDPE/PU blends, the samples were prepared according to ASTMD638 to measure with a universal testing machine (UTM, Instron 5982, Norfolk, MA, USA) with a load range of 100 kN and a crosshead speed of 5 mm/min. In addition, the flexural modulus of the samples (80 mm × 10 mm) was measured using the UTM with a range load of 10 kN, a crosshead speed of 50 mm/min, and a span distance of 48 mm.

The thermal shrinkage was confirmed as follows: the samples were prepared to a size of 4 cm × 4 cm and placed in an oven for 1 h at 150 °C and 180 °C to record their sample size. The thermal shrinkage was calculated with the following equation:Thermal shrinkage (%) = [1 − (T_a_/T_b_)] × 100,
where T_a_ and T_b_ represent the thermal shrinkage after and before the measured area, respectively.

## 3. Results and Discussion

### 3.1. Preparation of the HDPE/PU Blends

When blending polymers with different phases (e.g., polar and non-polar), they are not uniformly mixed, and phase separation occurs to produce materials in a rough and unstable state, and materials are prepared with weak mechanical and thermal properties [30,31]; therefore, to compensate for this limitation, it is possible to stably and uniformly blend the polymer with compatibilizers [32,33]. The compatibilizer is added to the immiscible polymers during blending and helps to stably mix the polymers [34,35,36]; therefore, we determined the optimal blending conditions by controlling the content of PE-g-MA for a uniform and stable blending of the non-polar polymer HDPE and polar polymer PU. PE-g-MA is a representative compatibilizer and has the role of enhancing the compatibility between polar and non-polar materials [37]. In addition, we were able to proceed rapidly with polymer crosslinking using TMPTMA. It is a viscous, colorless, and/or tan liquid used as a crosslinking agent, reactive diluent, and chemical intermediate [38,39]. Because TMPTMA is useful for low volatility and fast curing reactions, it has the characteristic of rapidly increasing gelation even with a small amount [40]; therefore, to establish the optimized blending conditions of HDPE and PU, different amounts of TMPTMA and PE-g-MA were mixed with HDPE/PU during the blending. After preparing sheets using the HDPE/PU blends with a 190 °C hot press, composites were obtained with excellent thermal and mechanical properties through electron-beam crosslinking (50, 75, and 100 kGy).

### 3.2. Characterization of the HDPE/PU Blends

Figure 1 shows the elemental composition analysis by EDS to confirm the uniform blending of the composite according to different amounts of PE-g-MA (3, 5, and 7 phr). Carbon (green) mainly identifies the carbon of the HDPE, and the oxygen (blue) and nitrogen (red) mainly identify the PU. In the merged image of F3, it can be seen that the oxygen and nitrogen are not uniformly distributed and agglomerated. It is considered that the PE-g-MA content (3 phr) was not optimal for uniformly blending HDPE and PU. On the other hand, in F5 and F7, the oxygen and nitrogen were uniformly distributed, so those blendings were effectively performed according to the different amounts of the compatibilizer.

Figure 2A shows the gel fraction according to the different amounts of PE-g-MA with an electron-beam irradiation dose of 100 kGy. The gel fraction was measured by Soxhlet extraction. Because 3 phr of TMPTMA was mixed in all samples, the gel fraction was confirmed to be approximately 68%. The gel fractions of F1, F3, F5, and F7 were 68%, 67.5%, 68%, and 68%, respectively. We confirmed that the different amounts of PE-g-MA did not affect the electron-beam crosslinking of the HDPE/PU blends. On the other hand, the mechanical properties were affected shown in Figure 2B,C. As shown in Figure 2B, the tensile strengths of F1 and F3 were 17.5 and 18 MPa, respectively; however, the tensile strengths of F5 and F7 were 22.5 and 23 MPa, respectively. This result shows the same trend as the EDS result in Figure 1. In F3, the amount of PE-g-MA at 3 phr was a limiting condition for the HDPE and PU to be uniformly blended, but F5 and F7 were uniformly blended; therefore, higher mechanical properties were obtained from F5 compared to F3. In the case of the tensile strain in Figure 2C, the values for F5 (560%) and F7 (560%) were slightly higher than those of F1 (500%) and F3 (510%), but there was no difference due to the excellent mechanical properties of PU.

Figure 3A shows optical images of the composites of the HDPE/PU blends at 150 and 180 °C prepared with different amounts of PE-g-MA. The optical images show that all the samples shrunk as the temperature increased from 23 °C to 180 °C, and the sheet for F1was conspicuously shrunken. On the other hand, although F5 and F7 were shrunk, they were maintained without being dramatically shrunken. The thermal shrinkage was calculated using the images (Figure 3B). The thermal shrinkage at 150 °C for F1, F3, F5, and F7 was 9.1, 6.9, 3.2, and 2.9%, respectively. In addition, the thermal shrinkage at 180 °C was 12.1, 9.1, 6.5, and 6.3%, respectively. F5 and F7 maintained their shape even at high temperatures. This result showed that the PE-g-MA content up to 5 phr had a role in uniformly blending the HDPE and PU. PU is used in electrical insulators, structural materials, and elastic fibers due to its excellent mechanical and thermal properties, such as abrasion resistance, cold resistance, and heat resistance [41,42,43]; therefore, there was little change in the thermal shrinkage of the composite due to the uniform blending of PU at a high temperature. These results confirmed the same results as our previous studies of PP blended styrene-grafted PU composites and HDPE/EVA/PU blends [13,22]. In addition, based on the blending conditions of F5, we wanted to confirm the blending effects according to the different amounts of PU.

As shown in the merged elemental image of F3 (Figure 1), it was observed that oxygen atoms of hydrophilic PU were aggregated without being dispersed in hydrophobic HDPE. On the other hand, it can be seen that oxygen atoms of hydrophilic PU are well dispersed in hydrophobic HDPE without aggregation in F5 and F7. These results indicate that high compatibility between hydrophobic HDPE and hydrophilic PU could improve the thermal/mechanical properties of the HDPE/PU. In this respect, the interfacial adhesion between HDPE and PU by working the PE-g-MA improved, resulting in superior thermal/mechanical properties compared to pristine HDPE; however, when the content of PE-g-MA is more than 5%, the thermal/mechanical properties of HDPE/PU blends could not be further improved; therefore, it is determined that the content of PE-G-MA in the HDPE/PU blend should be 5% or less in consideration of economic efficiency.

As shown in Figure 4, the elemental composition of the HDPE/PU blends was confirmed according to different amounts of PU (10; 20; 30 phr). As shown in the images of F5 and F7 in Figure 1, oxygen and nitrogen were uniformly distributed in all the samples. Based on the above results, PU 20 was prepared with the polymer and chemical additive concentration of F5. Even if the PU content was changed to 10 or 30 phr, the 5 phr of PE-g-MA effectively induced HDPE and PU blending. This result shows PU can stably act on the miscibility so that two polymers with different phases can be effectively blended for different amounts of PE-g-MA, thus, it is a compatibilizer.

Figure 5A shows the gel fraction according to the different amounts of PU by electron-beam irradiation of 100 kGy. The gel fraction of the HDPE without PU was confirmed to be 56%, and the HDPE/PU blends containing 10, 20, and 30 phr of PU had a gel fraction of 67, 68, and 70%, respectively. The gel fraction of PU 20 was increased by 12% compared to PU 0. PU and HDPE were crosslinked by electron-beam irradiation. In addition, to confirm the gel fraction according to the electron-beam irradiation dose, we additionally exposed PU 20 to electron-beam irradiation doses of 50 and 75 kGy. The gel fraction of PU 20–50 was 62% and PU 20–75 was 65%, as shown in Figure 5B. Based on this result, as the electron-beam irradiation dose was increased, the gel fraction increased. Figure 5C shows the mechanical properties of the composites by electron-beam irradiation. The tensile strengths of PU 0, PU 10, PU 20, and PU 30 without irradiation were 23, 22, 22.5, and 22 MPa, respectively. The tensile strengths of PU 0, PU 10, PU 20, and PU 30 with electron-beam irradiation at 100 kGy were 25, 23.5, 23 and 22.5 MPa, respectively. The tensile strength of the composite irradiated with 100 kGy was slightly increased compared to the composite that was not irradiated. On the other hand, large differences were confirmed in the tensile strain of the crosslinked composite by electron-beam irradiation. As shown in Figure 5D, the tensile strains of PU 0, PU 10, PU 20, and PU 30 without irradiation were 510, 550, 560, and 570%, respectively. The tensile strains of PU 0, PU 10, PU 20, and PU 30 with an electron-beam irradiation dose of 100 kGy were 110, 380, 420, and 450%, respectively. In the case of PU 0, there was a difference in the tensile strain of approximately 400% because it did not contain any PU, and only the HDPE was crosslinked. On the other hand, as the PU content increased, the difference in the tensile strain decreased compared to the composite not irradiated with the electron beam. This result is because the mechanical strength of the HDPE/PU blends is improved by the excellent mechanical properties of PU. Figure 5E shows the flexure strength of the different PU concentrations with electron-beam irradiation. The flexural modulus is a measure of how much a material can be bent without being permanently deformed and/or destroyed, and it represents the ratio of the stress applied to the deformation occurring in a bending test [44]. The flexural modulus of PU 0, PU 10, PU 20, and PU 30 without irradiation was 980, 660, 630, and 590 MPa, respectively. The flexural modulus of PU 0, PU 10, PU 20, and PU 30 with an electron-beam irradiation dose of 100 kGy was 1050, 805, 670, and 620 MPa, respectively. The flexural modulus showed a tendency opposite to the tensile strain, which shows that the mechanical properties were improved as the PU concentration was increased.

Figure 6A shows the thermal shrinkage results of PU 0, PU 10, PU 20, and PU 30 without electron-beam irradiation. In the case of PU 0, it was confirmed through optical images that the shape of the sheet rapidly shrunk as the temperature was increased from 23 to 180 °C. In addition, PU 10, PU 20, and PU 30 containing PU exhibited thermal shrinkage as the temperature was increased to 180 °C, but the thermal shrinkage was lower than that of PU 0. The thermal shrinkage of PU 0, PU 10, PU 20, and PU 30 without electron-beam irradiation at 150 °C was 9.8, 4.9, 3.2, and 2.9%, respectively. In addition, the thermal shrinkage of PU 0, PU 10, PU 20, and PU 30 without electron-beam irradiation at 180 °C was confirmed to be 12, 7, 6.5, and 6.3%, respectively. Figure 6B shows the optical images and shrinkage of PU 0, PU 10, PU 20, and PU 30 with an electron-beam irradiation dose of 100 kGy. In the case of PU 0, the thermal shrinkage was similar to PU 0 without electron-beam irradiation. On the other hand, the optical images of PU 10, PU 20, and PU 30 containing PU showed they maintained their shape even when the temperature was increased to 180 °C. The thermal shrinkage of PU 0, PU 10, PU 20, and PU 30 with an electron-beam irradiation dose of 100 kGy at 150 °C was 4.2, 1.2, 0.8, and 0.8%, respectively. At 150 °C, the PU 20 and PU 30 showed almost no shrinkage, evident by the optical images and thermal shrinkage. In addition, the thermal shrinkage rates of PU 0, PU 10, PU 20, and PU 30 with an electron-beam irradiation dose of 100 kGy at 180 °C were confirmed to be 5.5, 2.5, 2, and 1.8%, respectively. The HDPE/PU blends prepared through the electron-beam induced crosslinking reaction had improved thermal properties compared to the non-irradiated blends. In addition, we confirmed that the thermal properties were further improved with the inclusion of PU. This result is because hydrophilic PU with excellent thermal stability has improved compatibility with hydrophobic HDPE.

We optimized the PE-g-MA content to 5 phr by confirming the mechanical and thermal properties according to the different amounts of compatibilizer through the electron-beam crosslinking reaction. Then, based on the conditions used for F5, the thermal and mechanical properties with different PU concentrations were confirmed according to electron beam irradiation. Finally, we measured the mechanical properties at different temperatures (125 and 150 °C). Figure 7A shows the tensile strength of the different PU concentrations at different temperatures. The tensile strengths of PU 0, PU 10, PU 20, and PU 30 at 125 °C were confirmed to be 21, 22.5, 22, and 22 MPa, respectively. The tensile strengths of PU 0, PU 10, PU 20, and PU 30 at 150 °C were confirmed to be 14, 21, 21.7, and 21.2 MPa, respectively. At 125 °C, similar results for the mechanical strength were obtained regardless of the PU amount. On the other hand, it was shown that the mechanical strength was improved by the inclusion of PU at 150 °C. This is believed to be due to the excellent thermal and mechanical properties of PU. In addition, the tensile strains of PU 0, PU 10, PU 20, and PU 30 at 125 °C were confirmed to be 480, 530, 540, and 550%, respectively. The tensile strains of PU 0, PU 10, PU 20, and PU 30 at 150 °C were 320, 530, 530, and 535%, respectively, shown in Figure 7B.

## 4. Conclusions

In this study, we developed PE/PU blends with new and excellent physical properties through improved compatibility and electron-beam crosslinking. In general, the crosslinking structure of HDPE formed by electron beam irradiation improved the thermal stability and strength, but its elongation rapidly decreased; however, the PE/PU blend with improved compatibility (PU 30) showed an elongation of about 450% while maintaining excellent strength (22.5 MPa), which was increased by about four times compared to the HDPE irradiated at 100 kGy. In particular, the thermal stability of PU 30 irradiated at 100 kGy at a high temperature (180 °C) was improved by more than six times compared to HDPE. These excellent properties of the PE/PU blends, through improved compatibility and electron-beam crosslinking, can be used in insulated cables, pipes, and building materials that have to withstand harsh external environments.

## Figures and Tables

**Figure 1 polymers-14-03607-f001:**
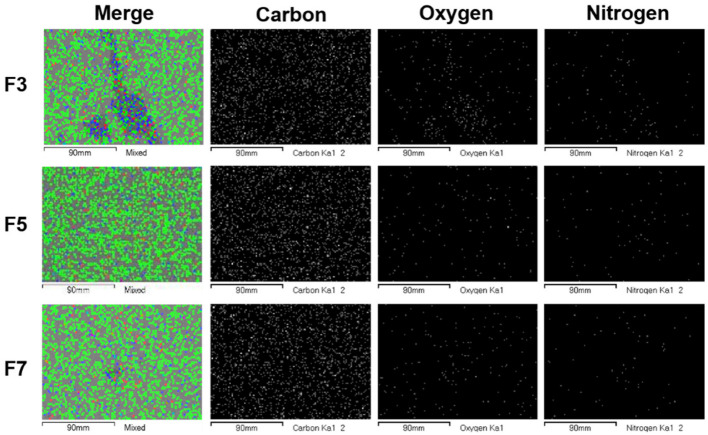
Elemental composition analysis of the HDPE/PU blends with PE-g-MA and TMPTMA by energy-dispersive X-ray spectroscopy. (Carbon: Green; Oxygen: Blue; Nitrogen: Red).

**Figure 2 polymers-14-03607-f002:**
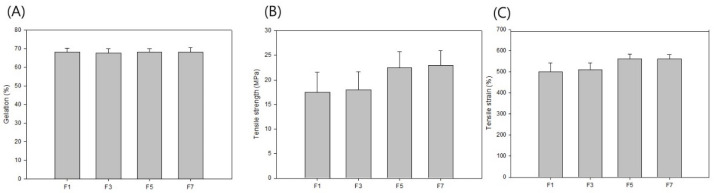
Physical and mechanical properties of HDPE/PU blends with compatibilizer at 100 kGy of radiation dose: (**A**) gel fraction; (**B**) tensile strength; (**C**) tensile strain.

**Figure 3 polymers-14-03607-f003:**
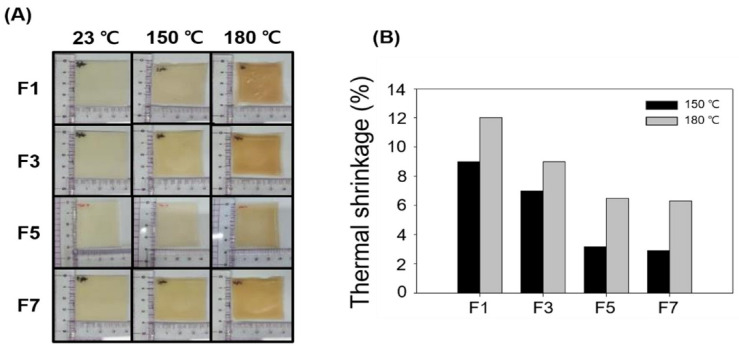
Thermal shrinkage of the HDPE/PU blends without irradiation at 150 and 180 °C for 1 h: (**A**) optical images; (**B**) reduction rate.

**Figure 4 polymers-14-03607-f004:**
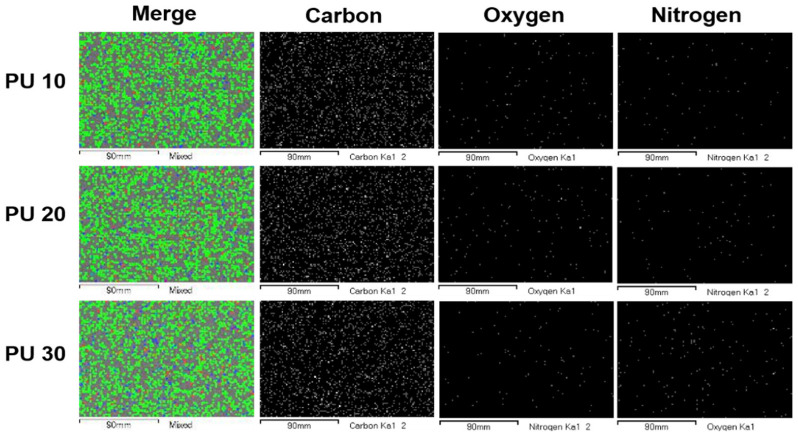
Elemental composition analysis of the HDPE/PU blends with different amounts of PU by energy-dispersive X-ray spectroscopy (Carbon: Green; Oxygen: Blue; Nitrogen: Red).

**Figure 5 polymers-14-03607-f005:**
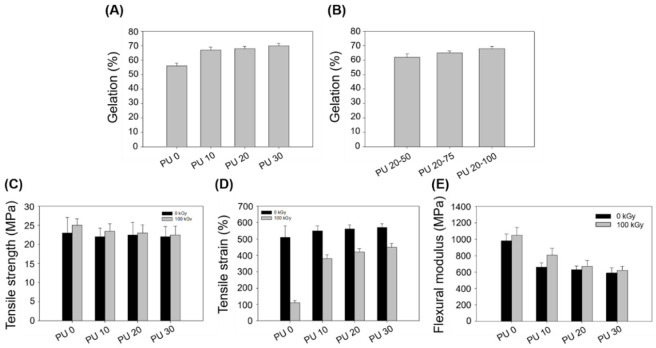
Gel fraction of the HDPE/PU blends: (**A**) Different amounts of PU with a radiation dose of 100 kGy. (**B**) Different radiation doses at 50, 75, and 100 kGy and mechanical properties of the HDPE/PU blends with a radiation dose of 100 kGy: (**C**) Tensile strength, (**D**) tensile strain, and (**E**) flexural modulus.

**Figure 6 polymers-14-03607-f006:**
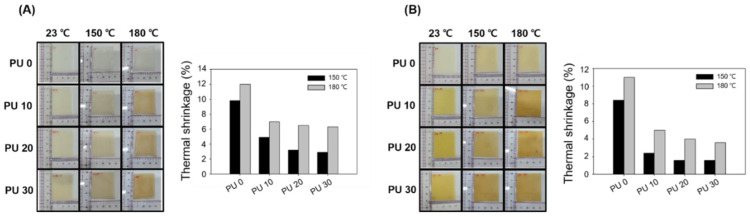
Thermal shrinkage of the HDPE/PU blends with different amounts of PU at 150 and 180 °C for 1 h: (**A**) 0 kGy and (**B**) 100 kGy.

**Figure 7 polymers-14-03607-f007:**
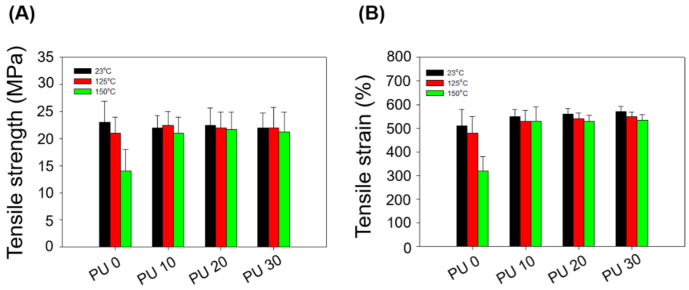
Mechanical properties of the HDPE/PU blends with different amounts of PU after thermal aging at 23, 125, and 150 °C for 2 h: (**A**) Tensile strength and (**B**) tensile strain.

**Table 1 polymers-14-03607-t001:** Chemical composition of the HDPE/PU blends with different amounts of the compatibilizer.

	F1	F3	F5	F7
HDPE	100 g	100 g	100 g	100 g
PU	20 phr	20 phr	20 phr	20 phr
PE-g-MA	1 phr	3 phr	5 phr	7 phr
TMPTMA	3 phr	3 phr	3 phr	3 phr

**Table 2 polymers-14-03607-t002:** Chemical composition of the HDPE/PU blends with different amounts of PU.

	PU 0	PU 10	PU 20	PU 30
HDPE	100 g	100 g	100 g	100 g
PU	-	10 phr	20 phr	30 phr
PE-g-MA	-	5 phr	5 phr	5 phr
TMPTMA	-	3 phr	3 phr	3 phr

## Data Availability

The data presented in this study are available on request from the corresponding author.

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
