# Peer review of "Optimization of the Physical Properties of HDPE/PU Blends through Improved Compatibility and Electron Beam Crosslinking"

_polymers, 2022, doi:10.3390/polym14173607_

Round 1

Reviewer 1 Report

The article is quite interesting, but needs a lot of improvement, especially when it comes to analyzing and correlating properties. The conclusions in the article are largely insufficient, it can be assumed that they are not there, so it is required to complete this important part of the article.

The literature analysis in the introduction is weak which needs to be improved.

Please describe in detail the process of making the discs.

Besides, I have some other comments and questions for the Authors.

Why is the term optimization used in the title, if there is nothing in the article about it?

line 22: 510-110, this notation is unfortunate, especially in the summary it should say how many percent the feature changes.

line 42-43, the names of materials should be correctly spelled with lowercase letters

line 96: What do the abbreviations in table F1-F10 mean?

line 115-117: what are the dimensions of the normalized samples? convert units to mm

line 164 should be MPa

line 168: it should be tensile ... written in lowercase, as indeed all other feature names, as well as the captions under figures 2, 3, etc.

Double numbering in the reference list

In summary, the work looks more like a research report than a scientific article, because the authors did not attempt to make conclusions supported by literature reports. Undoubtedly, this aspect should be completed.

Author Response

All comments and suggestions are greatly appreciated by authors since these suggestions and comments help us improve this manuscript. We have revised the manuscript carefully according to Reviewer's comment.

Point 1.

The conclusions in the article are largely insufficient, it can be assumed that they are not there, so it is required to complete this important part of the article.

Response 1:Thanks very much for your valuable opinion. A cross-linking structure of HDPE is formed by electron beam irradiation to increase its thermal stability and strength, but its elongation is rapidly decreased. In particular, the elongation of HDPE irradiated at 100 kGy was about 110%, which was decreased about 5 times compared to unirradiated HDPE (510%). However, the HDPE/PU blend with improved compatibility (PU 30) showed an elongation of about 450% while maintaining excellent strength (22.5 MPa), which was increased by about 4 times compared to the HDPE irradiated at 100 kGy. In particular, the thermal stability of PU 30 irradiated at 100 kGy at a high temperature (180 °C) was improved more than 6 times compared to the HDPE. The interfacial adhesion between HDPE and PU by working the PE-g-MA improved, resulting in superior thermal/mechanical properties compared to pristine HDPE. However, when the content of PE-g-MA is more than 5%, the thermal/mechanical properties of HDPE/PU blends could not be further improved. Therefore, it is determined that the content of PE-G-MA in HDPE/PU blend should be 5% or less in consideration of economic efficiency. Therefore, it is possible to develop HDPE/PU blends with new and excellent physical properties by improving the compatibility and using electron beam crosslinking technology.

Point 2.

The literature analysis in the introduction is weak which needs to be improved.

Response 2 The introduction part, we have revised this paper focusing on the reason for preparing the HDPE/PU blend.

“Polyolefin-based polymer resins are widely used in daily life because they have excellent mechanical strength, chemical resistance, and processability. However, these polymers have low thermal stability, impact strength, flame retardancy, and cold resistance. Polymer blending technology has been widely used to overcome these weaknesses of the homopolymer”:

Point 3.

Why is the term optimization used in the title, if there is nothing in the article about it?

Response 3: Thanks very much for your valuable opinion. As shown in Fig. 1 F3, it was observed that oxygen atoms of hydrophilic PU were aggregated without being dispersed in hydrophobic HDPE. On other hand, it can be seen that oxygen atoms of hydrophilic PU are well dispersed in in hydrophobic HDPE without aggregation aggregation in F5 and F7.  The interfacial adhesion between HDPE and PU by working the PE-g-MA improved, resulting in superior thermal/mechanical properties. However, when the content of PE-g-MA is more than 5%, the thermal/mechanical properties of HDPE/PU blends could not be further improved. Therefore, it is determined that the content of PE-G-MA in HDPE/PU blend should be 5% or less in consideration of economic efficiency.

Point 4.

line 22: 510-110, this notation is unfortunate, especially in the summary it should say how many percent the feature changes.

Response 4: Thanks very much for your valuable opinion. We have revised this sentence as follows. “In particular, the elongation of HDPE irradiated at 100 kGy was about 110%, which was decreased about 5 times compared to unirradiated HDPE (510%).”

Point 5.

line 42-43, the names of materials should be correctly spelled with lowercase letters

Response 5: Thanks very much for your valuable opinion. We have revised this sentence as follows. “Thus, polymers widely used in polymer blending include polypropylene (PP), polystyrene (PS), polyurethane (PU), and polyethylene (PE)”

Point 6.

line 96: What do the abbreviations in table F1-F10 mean?

Response 6:Thanks very much for your valuable opinion. The F1-F7 were chosen to be easy to read without any special meanings.

Point 7.

line 115-117: what are the dimensions of the normalized samples? convert units to mm

Response 7: Thanks very much for your valuable opinion. We have converted the unit for cm to mm.

“the flexural modulus of the samples (80 x 10 mm2) was measured using the UTM with a range load of 10 kN, a crosshead speed of 50 mm/min, and a span distance of 48 mm.”

Point 8.

line 164 should be MPa

Response 8: Thanks very much for your valuable opinion. We have revised this sentence as follows.

“ However, the tensile strengths of F5 and F7 were 22.5 and 23 MPa, respectively”

Point 9.

line 168: it should be tensile ... written in lowercase, as indeed all other feature names, as well as the captions under figures 2, 3, etc.

Response 9: Thanks very much for your valuable opinion. We have revised this sentence and caption of figures 2, 3 as follows.

“In the case of the tensile strain in Fig. 2C, the values for F5 (560%) and F7 (560%) were slightly higher than those of F1 (500%) and F3 (510%), but there was no difference due to the excellent mechanical properties of PU.”

“Figure 2. Physical and mechanical properties of HDPE/PU blends with compatibilizer at 100 kGy of radiation dose: (A) gel fraction. (B) tensile strength. (C) tensile strain.”

“Figure 3. Thermal shrinkage of the HDPE/PU blends without irradiation at 150 and 180 °C for 1 h: (A) optical images and (B) reduction rate”

Point 10.

Double numbering in the reference list.

Response 10: Thanks very much for your valuable opinion. We have double-checked the number of reference

Point 11.

In summary, the work looks more like a research report than a scientific article, because the authors

did not attempt to make conclusions supported by literature reports. Undoubtedly, this aspect should be completed

Response 11: Thanks very much for your valuable opinion. The reason for the result’s behaviors is discussed more deeply.

“As shown in the merged elemental image of F3 (Fig. 1), it was observed that oxygen atoms of hydrophilic PU were aggregated without being dispersed in hydrophobic HDPE. On other hand, it can be seen that oxygen atoms of hydrophilic PU are well dispersed in in hydrophobic HDPE without aggregation in F5 and F7.  These results indicate that high compatibility between hydrophobic HDPE and hydrophilic PU could positively improved on the thermal/mechanical properties of the HDPE/PU. In this respect, the interfacial adhesion between HDPE and PU by working the PE-g-MA improved, resulting in superior thermal/mechanical properties compared to pristine HDPE. However, when the content of PE-g-MA is more than 5%, the thermal/mechanical properties of HDPE/PU blends could not be further improved. Therefore, it is determined that the content of PE-G-MA in HDPE/PU blend should be 5% or less in consideration of economic efficiency.”.

Reviewer 2 Report

In this manuscript, the authors improved the thermal and mechanical properties of the high-density polyethylene (HDPE) / polyurethane (PU) blends. In addition, they studied the effect of electron-beam crosslinking on the blend. A cross-linking structure of HDPE is formed by electron beam irradiation to increase its thermal stability and strength, but its elongation is rapidly decreased. In particular, the elongation of HDPE irradiated at 100 kGy decreased about 5 times compared to unirradiated HDPE (510% → 110%). However, the HDPE/PU blend with improved compatibility (PU 30) showed an elongation of about 450% while maintaining excellent strength (22.5 MPa), which was increased by about 4 times compared to the HDPE irradiated at 100 kGy. In particular, the thermal stability of PU 30 irradiated at 100 kGy at a high temperature (180 °C) was improved more than 6 times compared to the HDPE. Therefore, it is possible to develop HDPE/PU blends with new and excellent physical properties by improving the compatibility and using electron beam crosslinking technology. These excellent properties of the PE/PU blend through improved compatibility and electron-beam crosslinking can be used in insulated cables, pipes, and building materials that have to withstand harsh external environments. This study is good and important to polymer blend subject to produced new polymer composite with excellent thermal and mechanical properties. The interpretations of the results need to be deeply and improved to discuss the reason for the result’s behaviors. The quantity and quality of the figures are appropriate. We believe that this research subject is promising for developing polymer blends with high mechanical compared to a homopolymer.

Summary: I recommend publishing this manuscript after considering my comments on the attached file.

Author Response

All comments and suggestions are greatly appreciated by authors since these suggestions and comments help us improve this manuscript. We have revised the manuscript carefully according to Reviewer's comment.

Point 1.

In this manuscript, the authors improved the thermal and mechanical properties of the high-density polyethylene (HDPE) / polyurethane (PU) blends. In addition, they studied the effect of electron-beam crosslinking on the blend. A cross-linking structure of HDPE is formed by electron beam irradiation to increase its thermal stability and strength, but its elongation is rapidly decreased. In particular, the elongation of HDPE irradiated at 100 kGy decreased about 5 times compared to unirradiated HDPE (510% → 110%). However, the HDPE/PU blend with improved compatibility (PU 30) showed an elongation of about 450% while maintaining excellent strength (22.5 MPa), which was increased by about 4 times compared to the HDPE irradiated at 100 kGy. In particular, the thermal stability of PU 30 irradiated at 100 kGy at a high temperature (180 °C) was improved more than 6 times compared to the HDPE. Therefore, it is possible to develop HDPE/PU blends with new and excellent physical properties by improving the compatibility and using electron beam crosslinking technology. These excellent properties of the PE/PU blend through improved compatibility and electron-beam crosslinking can be used in insulated cables, pipes, and building materials that have to withstand harsh external environments. This study is good and important to polymer blend subject to produced new polymer composite with excellent thermal and mechanical properties. The interpretations of the results need to be deeply and improved to discuss the reason for the result’s behaviors. The quantity and quality of the figures are appropriate. We believe that this research subject is promising for developing polymer blends with high mechanical compared to a homopolymer.

Response 1: Thanks very much for your valuable opinion. The reason for the result’s behaviors is discussed more deeply.

“As shown in the merged elemental image of F3 (Fig. 1), it was observed that oxygen atoms of hydrophilic PU were aggregated without being dispersed in hydrophobic HDPE. On other hand, it can be seen that oxygen atoms of hydrophilic PU are well dispersed in in hydrophobic HDPE without aggregation in F5 and F7.  These results indicate that high compatibility between hydrophobic HDPE and hydrophilic PU could positively improved on the thermal/mechanical properties of the HDPE/PU. In this respect, the interfacial adhesion between HDPE and PU by working the PE-g-MA improved, resulting in superior thermal/mechanical properties compared to pristine HDPE. However, when the content of PE-g-MA is more than 5%, the thermal/mechanical properties of HDPE/PU blends could not be further improved. Therefore, it is determined that the content of PE-G-MA in HDPE/PU blend should be 5% or less in consideration of economic efficiency.”

Reviewer 3 Report

In this work we developed PE/PU blends with new and excellent physical properties through improved compatibility and electron-beam crosslinking. Authors have shown an enhancement of physical properties bay blending to polymers. Overall this is a comprenhensive manuscript and this work will be in the field of interest of researchers community, for that I recommended the publication of this paper after these revisions .

1/ The introduction should be improved specially in the end what is the application of this blendes polymers? Were we can used this blending polymers?

2/ In Figure 1 why the composition of F10 is not presented? I don’t feel that therese a difference between F5 and F7. What is the optimal composition?

3/ In Figure 2 the same Remarque why F10 not presented or not measured to better compared all blended compound? As shown in Figure 2 Therese no significant change between F1 and F3 as well as between F5 and F7? This is should be more clarified? 

4/ As looking in figure 4 could you say that PU or PE-g-Ma is comported as a good disperse? What can you say about the effect of crosslinked phenomena is affected or it have an effect for these compounds (PU 10, PU20 PU30..) ?

5/ These papers should be cited

Ltayef, M.Almoneef, M.M.Taouali, W. M. Mbarek.Alimi, K. Polymers, (2022), 14(3), 513

Abbassi, F. M. Mbarek.Almoneef, M.Alimi, K. Polymers, 2021, 13(17), 2902

M. Mbarek, F. Abbassi, K.Alimi, Journal of Molecular Structure 1120 (2016) 125-131

F. Abbassi, M. Mbarek, K. Alimi, Journal of Physics and Chemistry of Solids 126, (2019), 274-279

Author Response

(The authors gave the same response as above.)

Round 2

Reviewer 1 Report

I would like to thank the Authors for comprehensive answers to the questions and comments I posed in the review, which I find satisfactory. I believe that in its current form, the article constitutes valuable scientific material that can be published.

Reviewer 3 Report

This paper can be published in Polymers journal in this form.  Authors have taken in account all my comments.